# The Long-Term Dietitian and Psychological Support of Obese Patients Who Have Reduced Their Weight Allows Them to Maintain the Effects

**DOI:** 10.3390/nu13062020

**Published:** 2021-06-11

**Authors:** Katarzyna Iłowiecka, Paweł Glibowski, Michał Skrzypek, Wojciech Styk

**Affiliations:** 1Department of Biotechnology, Microbiology and Human Nutrition, University of Life Sciences in Lublin, 20-704 Lublin, Poland; katarzyna.ilowiecka1@gmail.com; 2Department of Clinical Dietetics, Medical University of Lublin, 20-093 Lublin, Poland; michal.skrzypek@umlub.pl; 3Institute of Psychology, The John Paul II Catholic University of Lublin, 20-950 Lublin, Poland; wojciech.styk@kul.pl

**Keywords:** obesity, obesity treatment, weight loss, weight loss maintenance, body composition, weight regain, energy-restricted diet

## Abstract

The role of post-therapeutic support after weight loss in obesity treatment is not fully understood. Therefore, weight maintenance after a successful weight loss intervention is not very common, especially in obese individuals. This randomized controlled study was conducted to explore the efficacy of following dietary and psychological support in a group of 36 obese individuals. Participants (22 women, 14 men aged 35.58 ± 9.85 years, BMI 35.04 ± 3.80 kg/m^2^) who completed a 12-month weight loss phase (balanced energy-restricted diet) were randomly allocated to receive 18-month support (SG) or no additional care (CG). The support phase included some elements of Ten Top Tips (TTT), cognitive behavioral therapy (CBT), motivational interviewing (MI) in combination with nutritional education and assessment of the level of physical activity. The primary outcome was the maintenance of anthropometric parameters at an 18-month follow-up. The secondary outcomes included evaluation of biochemical parameters and single nucleotide polymorphisms (SNPs) in genes connected with obesity. A comparison of SG vs. CG after a 30-month period of the study revealed significant differences in weight changes (−3.83 ± 6.09 vs. 2.48 ± 6.24 kg), Body Mass Index (−1.27 ± 2.02 vs. 0.72 ± 2.12 kg/m^2^), visceral adipose tissue (−0.58 ± 0.63 vs. 0.45 ± 0.74 L), and waist circumference (−4.83 ± 4.05 vs. 1.83 ± 5.97 cm). Analysis of SNPs (rs9939609 FTO, rs987237 TFAP2B, and rs894160 PLIN1) provided further insight into the potential modulating effect of certain genotypes on weight loss and maintenance and extended the knowledge of the potential benefits of personalized medicine. Post-therapeutical support in current clinical practice may increase the chances of long-term weight loss maintenance in obesity treatment even in patients with a genetic predisposition to excessive weight.

## 1. Introduction

According to the World Health Organization (WHO), near to two billion adults were overweight in 2016. Of these, more than 650 million were obese [1]. Obesity is characterized by excessive fat deposition, which is a consequence of a long-term energy surplus. The factors responsible for this state are primarily seen in excessive energy consumption, reduced energy expenditure, or both at the same time. Common (multifactorial) obesity most likely results from a concerted interplay of genetic and environmental factors [2]. Although many studies try to explain the correlations between various single nucleotide polymorphisms (SNPs) and obesity [3,4], a broader analysis, including the time of weight maintenance after the weight loss process, has been rarely carried out until now.

Currently, several indicators are used to diagnose obesity. The body mass index (BMI), which is defined as the weight in kilograms divided by the height in meters squared (kg/m^2^), is the most common measure of obesity. Besides the BMI, the most important parameters related to obesity are waist circumference (WC) and percentage of fat mass (FM%). As indicated by the WHO, a WC >80 cm in European women and >94 cm in European men is connected with an increased risk of metabolic disorders, and the risk is increased significantly with a WC >88 cm in women and >102 cm in men [5]. According to the WHO guidelines [6], obesity occurs when the body fat content is above 25% in men and 35% in women. Studies connected with obesity measures also incorporate a variety of more advanced techniques to estimate body composition, including bioimpedance analysis (BIA), which is a fast, simple, and non-invasive method of determining the value and percentage of fat mass, fat-free mass (FFM), visceral adipose tissue (VAT), or the body hydration level [7]. Another characteristic of obesity is changes in the biochemical blood parameters, e.g., insulin, hematocrit, triglycerides, low-density lipoprotein (LDL), very low-density lipoprotein (V-LDL), or alanine aminotransferase (ALT) [8]. To date, however, there are no reports in the research literature on not only changes in bodyweight after the weight loss process but also the extensive body composition and biochemical parameters of the blood, which are one of the most important indicators of metabolic health and nutritional status.

Weight reduction in overweight patients, especially under a dietitian’s supervision, is not a substantial scientific challenge. The Achilles heel of all weight loss therapies is the maintenance of achieved effects. Weight regain remains the most common long-term outcome of most interventions. Some estimates suggest that this problem will concern 80% of obese patients undergoing treatment [9]. On average, patients regain much of their weight within two years after discontinuation of treatment. A lot of metabolic and psychological aspects have to be taken into account to realize the problem of weight loss maintenance. Primly, the weight loss process leads to metabolic adaptations that promote repeat weight gain [10,11]. Metabolic mechanisms include a reduction of the resting metabolic rate, decreased production of some bioactive hormones e.g., of thyroid, reducing the amount of heat produced, changes in the concentration of leptin in the body, and lowering of muscle performance (burning fewer calories per unit of work) [12,13]. Mentioned above actions are aimed to save energy during energy deficits. Moreover, during a long-term energy restriction hormonal signals are produced, which increase the feeling that energy intake should increase. Levels of hormones involved in the regulation of appetite like cholecystokinin, ghrelin, peptide YY, insulin, leptin, and gastric inhibitory polypeptide change in response to the long weight loss process, causing increasing food demand, and intensified perception of hunger [10]. In addition, there are reports suggesting that some of mentioned above changes can persist for several years, even after the weight loss process is over [14]. Also, psychological aspects (e.g., high-stress levels [15]) and persisting obesogenic environments contribute to weight regain [16,17]. The current evidence is compatible with the notion that high motivation is one of the key predictors of successful weight loss and maintenance [18,19]. Additionally, over time, individuals generally have a gradual decrease in overall adherence to the initial dietary prescription. This typically results in a slowly increasing return to the pretreatment level of calorie intake. A gradual return to the eating pattern or pretreatment macronutrient profile and physical activity level regress over time. Some research suggests that, after the first month of dieting, self-rated adherence to dietary recommendations and physical activity levels decrease each month [13,20]. Once the target is reached, patients usually stop visiting a dietitian and most often gain weight again. The same pattern is seen by those who lose weight without professional support. The maintaining weight loss factors include high physical activity levels and continued consumption of low-energy meals, including meal replacements [21]. 

Although it is widely believed that weight regain is inevitable, there is some evidence to suggest that it is possible to maintain weight loss for a long time [22]. In this context, it is emphasized that compliance with the entire set of behaviors is required in order to prevent an energy surplus [23]. These changes include behaviors such as eating habits, meal structure, food choices, alcohol and sweets consumption, calorie counting, an adequate amount of sleep, stress management, or leisure activities. There is also scientific evidence to suggest that people who successfully maintain their bodyweight have developed a habit of exercising for 30–60 min a day or frequently engaging in other activities that require vigorous physical activity [24,25]. On the other hand, people with weight gain prefer a less active lifestyle [26]. Although changing these behaviors is referred to as a single lifestyle modification, it affects many areas of daily life. Therefore, it is observed that even after several years of successful weight maintenance, this process is still a challenge [27,28]. Few studies have been conducted with a multidimensional maintenance design where the individual reduces weight using a balanced approach followed by randomization to weight loss maintenance or control interventions [29]. Many randomized controlled trials of weight maintenance focus on a retrospective follow-up of individuals who participated before in an active period of weight loss [30,31,32,33]. The phase of maintenance concentrates mainly on keeping low energy density meals in the diet or addition of some nutrients [34,35], only psychological programs for behavior change [36], increasing the level of physical activity [37], or all of the assumptions mentioned above at once [38]. The weight loss maintenance phase described in the present study is based on dietary and psychological support (Ten Top Tip methods, some elements of integrating cognitive behavioral therapy (CBT), and motivational interviewing (MI), which has not been investigated previously.

Due to the increasing prevalence of obesity, development of practical treatment approaches has been identified as a research and population health priority [39]. To the best of our knowledge, there is no answer to the question of the effects of post-therapeutic care on changes in relevant determinants of successful bodyweight reduction, i.e., the extensive body composition analysis or biochemical blood parameters with simultaneous analysis of various obesity-related SNPs. Identification of these correlations can help develop a helpful strategy after the weight loss process.

To address the information presented above, the aim of the study was to assess whether comprehensive post-therapeutic care can help in long-term maintenance of effects achieved during the bodyweight reduction process, i.e., upgraded body composition and blood analysis results. Given the prevalence of obesity, it seems advisable to develop standards for clinicians and dietitians specifying interventions targeted at maintenance of weight loss effects.

## 2. Materials and Methods

### 2.1. Study Design

#### 2.1.1. Phase 1 of the Study—Weight Loss Process

In the beginning, 50 obese patients from the Lublin region, Poland, were recruited to the 3-year intervention study. The recruitment channels included social media and University notice boards. The first phase (Ph1) of the study assumed an annual weight loss process under supervision of a qualified dietitian. The dietary intervention was designed based on the Polish Dietetic Association guidelines [40]. The energy deficit used was in the range of 500–800 kcal/d. The distribution of energy from macronutrients was 15–25%, 25–35%, and 45–60% from proteins, fats, and carbohydrates, respectively. In this phase, special attention was paid to personalization of the nutritional plan and an individual approach to the patient. The details of the intervention have been described in our previous paper [41]. Briefly, a nutrition interview was conducted, and five-day food diaries prepared by the participants were analyzed to compose a personal diet for each patient. The diet plan met all the assumptions (qualitative and quantitative) of a healthy, balanced diet, and included 4 or 5 meals per day. Participants were also encouraged to achieve the goal of 75 min of high-intensity physical activity or 150 min of moderate activity per week [42]. During the follow-up visits (at least 10), anthropometric measurements were performed and the diet plans were modified. The follow-up visits were monthly and usually on the same day each month to minimize the impact of different phases of the menstrual cycle on the obtained results. The participants had unlimited access to support (online or telephone) from a qualified dietitian if needed.

#### 2.1.2. Phase 2 of the Study—Post-Therapeutic Care

After a 12-month weight loss process, 36 participants joined the second phase (Ph2) lasting 18 months. The pair-matching strategy was applied during division of the participants into two groups: supported (SG) and control (CG). After Ph1, CG did not receive any dietary care. In contrast, SG was invited to participate in group and individual meetings conducted by a qualified dietitian and a psychologist. The main assumptions in this stage comprised regular dietary education, in particular focusing on the most practical information of balanced nutrition, and psychological support. Based on the acquired knowledge, the participants were expected to become independent in weight loss maintenance without applying a strict nutritional plan. Teaching them correct eating habits was a priority in this phase. Ph2 was based on regular (at least once a month) group meetings where patients had the opportunity to acquire new knowledge and exchange experience. If applicable, there was also a possibility of individual sessions with a psychologist or dietitian (without a chance of obtaining a diet plan). The topics discussed during Ph2 included the assumptions of the Ten Top Tips (TTT) method, “which encouraged daily repetition of ten behaviors proposed to create a negative energy balance and subsequent weight loss. The behaviors included: (1) keep to a meal routine, (2) eat reduced-fat foods, (3) walk 10,000 steps a day, (4) pack a healthy snack, (5) check food labels, (6) watch portion sizes, (7) stand up for 10 min in every hour, (8) choose low-calorie drinks, (9) be mindful when eating, and (10) eat five portions of fruit and vegetables a day” [43]. Additionally, the discussed topics were related to the basic guidelines of balanced nutrition, rational purchasing, healthy substitutes for popular products, knowledge of portion size, valuable tips for holidays and family celebrations, or most common mistakes during weight loss. The psychological part was based on psychoeducation and some elements of integrating cognitive behavioral therapy (CBT) and motivational interviewing (MI), which might be effective methods in obesity reduction [44,45]. CBT is traditionally recognized as the best-established long-term perspective treatment for obesity [46]. Ph2 used some CBT assumptions, i.e., self-monitoring goal setting, stimulus control, contingency management, skills for increasing social support, problem-solving, and relapse prevention. MI aims to encourage patients into action by identifying discrepancies between their current behavior and desired goals. The principles and methods of MI address issues associated with ambivalence about behavior change, including decreased confidence and low self-efficacy [45]. The specific MI strategies that were used include enhancing decisional balance, exploring values, exploring importance and faith, and identifying specific measurable and feasible objectives [47]. The psychological part was also focused on minimizing stress, psychological factors of obesity, or support from the environment.

The COVID-19 pandemic made it challenging to conduct an experiment, and the lockdown made it impossible to continue direct meetings. For this reason, Ph2 was continued via the Internet. The participants were receiving thematic newsletters or nutrition pamphlets related to weight control. They were also communicating and supporting each other (e.g., by exchanging photos of meals, recipes, or pieces of information about the effects) using a popular communicator. A simple scheme of the interventions used in Phase 1 and Phase 2 can be found in the Appendix A.

### 2.2. Test Group

The investigation was conducted in the Dietitian Service (University of Life Sciences in Lublin, Poland) from March 2018 to February 2021 on 36 volunteers. Among these 61% (n = 22) were Caucasian women, and 39% (n = 14) were Caucasian men. The age range of the participants was 21–49 years. All participants gave their informed consent for inclusion before the study. In Ph2, the basis of the allocation consisted in matching one CG member to an SG member, given the three leading variables necessary for the study: BMI value, age and sex. Each member of the pair was randomly assigned to the CG or SG group. This process was performed by one member of our team (K.I.) using sealed envelopes. The details of the exclusion and inclusion criteria (based mainly on AACE/ACE classification [48]) have been described in our previous paper [41]. The study was conducted in accordance with the Declaration of Helsinki, and the Ethics Committee of the Medical University of Lublin approved the protocol. Decision number: KE-0254/180/2019. This study followed the guidelines set out in the Consolidated Standards of Reporting Trials (CONSORT) [49].

### 2.3. Outcome Measurements

Anthropometric measurements of all participants were made at the starting point, after 12 months of Ph1, and after 18 months of Ph2. Moreover, body composition analysis was performed in the SG group in Ph1 every month. Anthropometric analyses were carried out according to the method described previously by Banach et al. [41]. Briefly, besides body height, mass and the WC, a SECA mBCA515 (seca GmbH & Co. KG., Hamburg, Germany) analyzer was applied for body composition analysis using the bioelectric impedance method (eight-point). In recent years, scientific literature has found the usefulness of BIA applied in the SECA mBCA 515 methodology (i.e., in a standing position, without electrodes), to assess body composition [50,51]. It should be mentioned that the disproportion between body mass and body conductivity lowers the accuracy of BIA in obesity. Kyle et al. [52] demonstrated that BIA is valid with BMIs up to 34 kg/m^2^. In morbid obesity, the most predictive equations cannot predict the static body composition and are not reproducible for individuals over time. Moreover, changes in anthropometric parameters (e.g., FFM, FM%, TBW) that the BIA can capture in prospective observation are limited to values less than 1.5–2.0 kg [52]. Despite these limitations, BIA is the most commonly used method of assessing body composition in dietary practice [53].

The level of physical activity was determined using a validated International Physical Activity Questionnaire (IPAQ)—A long version. This research tool consists of 27 questions grouped into five independent parts. These questions apply to all possible physical activities undertaken within the last seven days. The first four parts include assessment of physical activity in different domains of everyday life (e.g., housework, job-related activities, sports, transportation, and recreational activities). The last part is focused on the amount of time spent sitting [54]. The minimum activity pattern to be classified as “sufficiently active” is any of the following criteria: “at least 20 min of vigorous-intensity activity per day for three or more days per week, at least 30 min of moderate-intensity exercise per day for five or more days per week, five or more days of any combination of walking, moderate-intensity or vigorous-intensity activities achieving a minimum total physical activity of at least 600 MET min/week” [55]. The metabolic equivalent task (MET) is defined as oxygen consumption at rest and is 3.5 mL O_2_ per kg of bodyweight per minute [56]. Individuals meeting at least one of these criteria would be defined as achieving the minimum recommended considered ”minimally active” [55]. The participants were asked to fill in the questionnaire during the meetings at the Dietitian Service. Data collection using IPAQ was an integral part of the study. Self-completion of the questionnaire took place at the same time as the BIA and blood analyses.

Fasting blood samples (morphology, total cholesterol (Total-chol), low-density lipoprotein (LDL), high-density lipoprotein (HDL), triglycerides (TG), aspartate transaminase (AST), alanine aminotransferase (ALT), thyroid-stimulating hormone (TSH), glucose level, and C-reactive protein (CRP) were collected and analyzed by an external diagnostic laboratory. Glucose was determined with the automatic enzymatic colorimetric method using hexokinase. TC, HDL-C, LDL-C, and TG were analyzed with the automatic enzymatic colorimetric method using commercial kits. TSH was measured using an enzyme immunoassay (Abbott) with an operating range of 0.01–100 mU/L, and CRP was measured using an enzyme-linked immunosorbent assay (ELISA).

Each participant was genotyped for three genes: FTO (variant rs9939609), TFAP2B (rs987237), and PLIN1 (rs894160). The identification was carried out by an external genetic laboratory.

During the last visit to the Dietitian Service, all participants were asked to complete an author’s questionnaire related to their participation in the study and the impact of the COVID-19 pandemic on the weight loss process and changing eating habits.

### 2.4. Statistical Analysis

The results were subjected to statistical analysis. We used the following statistical tests: the Shapiro-Wilk test to evaluate the normality of the distribution for continuous parameters. The Student’s test to compare two average values, or the non-parametric Wilcoxon test to compare values without a normal distribution. The ANOVA test with repeated measures was used to compare the three average values. Values of the continuous parameters were presented as the mean value and standard deviation, and the categorical ones were shown as the number and percentage. A significance level of *p* < 0.05 was assumed to indicate statistically significant differences. Analyzes were performed by STATISTICA 13.3 computer software (StatSoft, Inc., Tulsa, OK, USA).

## 3. Results

At the end of Ph2, 36 participants completed the study and were included in the statistical analysis (Figure 1). No adverse effects of the implementation of the reduction diet were reported during the experiment. The descriptive characteristics of the subjects assessed at the beginning of the study are shown in Table 1.

At the beginning of the study, the mean age of the participants was 35.58 ± 9.85 years, and their height was 1.72 ± 0.10 m. Data in Table 1 indicate that the participants did not achieve the best results at the end of Ph1. After the initial rapid achievement of better anthropometric measurement values, the results began to change adversely in the last six month of Ph1 but did not return to the starting point.

The diet’s energy value in Ph1 was approximately 1550 kcal/d for women and 2100 kcal/d for men. Depending on the individual case, the energy deficit was 500–800 kcal/d, which meets the Polish guidelines of a balanced weight loss diet [40]. Most of the analyzed micronutrients met the nutrient requirements of the Recommended Dietary Allowances (RDA) or Adequate Intake (AI) standard. 

Table 2 presents the results of changes in blood biochemical parameters in the two groups at the beginning and end of the 12-month weight loss intervention.

As expected, during the weight loss phase, the levels of glucose, TG, and TSH decreased, although no significant (*p* < 0.05) differences were found and the effect size of obtained results was small or medium. A significant increase in total cholesterol and LDL was observed, which may be a transition state caused by a change in lipoprotein metabolism during weight loss. During the progression of weight loss, adipose cholesterol, typically stored in fatty tissue, gets into the bloodstream, causing changes in the lipidogram. This effect is not permanent, and cholesterol levels should decrease when weight stabilizes [58,59]. Besides, in Poland, hypercholesterolemia is generally widespread. This fact was demonstrated in the WOBASZ II study, where lipid disorders, including excess levels of LDL and TG in a representative sample, were reported in 61% of women and men aged 20–74 years [60]. The plasma liver damage markers, i.e., the ALT and AST activities, were significantly reduced due to caloric restriction. This is especially important in non-alcoholic fatty liver disease (NAFLD). Most patients with NAFLD are asymptomatic. The disease is typically suspected based on elevated ALT and AST levels and other clinical and biochemical features or incidental findings during abdominal ultrasonography [61]. To assess NAFLD, we analyzed medical records in the process of exclusion of complicated obesity. The resolution of fatty liver was observed in three patients during Ph1. The levels of leukocytes, erythrocytes, hemoglobin, and platelets confirmed that our patients were rather healthy individuals and weight reduction generally did not affect these parameters. Similar results were obtained in the further part of the study (Table 3 and Table 4).

Table 3 presents the results of anthropometric and biochemical measurements in the two groups (SG and CG) at the beginning and end of the 18-month support intervention. 

The post-therapeutic support phase did not contribute to significant changes in the anthropometric parameters in the SG group. However, the overall weight loss was maintained, although no further weight loss occurred. Over 18 months, the patients maintained the effects achieved in Ph1, and effect size of obtained results was small (range 0.01–0.28) for anthropometric parameters and small or medium for biochemical parameters (range 0.02–0.59). In turn, a significant increase in bodyweight, BMI, fat mass, VAT, and WC occurred in the CG group. In most parameters, the adverse changes were twice as high as in SG. The discontinuation of contact with a dietitian caused an increase in obesity. Moreover, effect size of results in CG was higher than in SG (range 0.10–0.47 for anthropometric parameters and 0.09–0.69 for biochemical parameters), which proves a greater severity of adverse changes. An increase in the fasting blood glucose level was recorded in both groups, and an increased ALT and CRP concentration was additionally detected in CG. The decrease in HDL may have been caused by the weight gain and reduced physical activity levels in both groups [62]. One of the causes of the increase in the glucose levels in both groups may have been the negative impact of the COVID-19 lockdown on participants’ food choices (Appendix A). In patients with pre-diabetes (glucose level ≥ 100 mg/dL), a general practitioner (GP) follow-up was recommended. The phase angle is an indicator of cellular integrity and has been proposed as a prognostic parameter for cell conditions. In healthy adults, the phase angle is usually in the range of 5.0–7.0°, and the value below 5.0° indicates malnutrition [63]. The present results indicate a satisfactory state of cells in the studied group. The weight loss phase and (or) weight maintenance did not change this indicator significantly.

Table 4 presents the results of anthropometric and biochemical measurements in the two groups (SG and CG) at the beginning and end of the 30-month study.

Table 4 summarizes the results in both groups from the beginning of the study to the 30-month assessment. Excessive accumulation of VAT leads to visceral obesity. As shown by Peine et al., higher values than 1.9 L for women and 3.8 L for men indicate excessively high levels [64]. At 30 months, the support group achieved several favorable effects from the beginning of the study, including bodyweight and BMI reduction and a decreased level of VAT and WC. This shows that the long-term care contributed to improvement of important anthropometric parameters (Figure 2). In contrast, in CG, there were no significant differences in most anthropometric parameters between 0 and 30 months. The exception was VAT, which was significantly less beneficial. In the case of biochemical parameters, an increase in glucose and total-chol was noticed. Only one positive effect was demonstrated for CRP. No significant changes in the complete blood count were noticed throughout the study, although obesity is a chronic inflammatory condition, and leukocytes have widely recognized associations with inflammatory conditions [65]. This study also proved that leukocytes, erythrocytes, hemoglobin, and platelets have a limited value as clinical markers of obesity. Effects-size in SG showed similar values for all measured parameters: range 0.03–0.37 (very small or small). In contrast, in CG a large effect size for glucose increase (0.9) was observed, and a medium effect size for the increase in total cholesterol, LDL, TG, and decrease in AST (0.50). The other values ranged from 0.02 to 0.22.

Genetic tests were made to assess the impact of single nucleotide polymorphisms associated with the ability to reduce bodyweight and maintain the achieved effects. The frequency distribution of SNPs connected with excessive bodyweight in the study group is presented in Table 5.

As shown in Table 5, the SNP distribution in SG and CG was similar. In the case of the FTO gene, most individuals were characterized by the presence of one risk allele (A). On the other hand, the most favorable GG variant most often appeared in relation to analysis of the PLIN1. Only 11% of a participants had GG variant in TFAP2B gene, which in most studies was negatively associated with obesity.

Table 6 shows the distribution of anthropometric variables according to the genotypes of the SNPs rs9939609, rs987237, and rs894160 in all participants in the 6th and 30th month of the study.

This analysis was carried out to evaluate whether individual SNPs can affect the weight loss process, weight regain, and other anthropometric parameters. Three genes were analyzed. FTO, whose polymorphisms are strongly associated with obesity and fat mass, exerts the greatest influence on BMI of all known genes [66]. PLIN1 is also associated with an excessive bodyweight risk and complications of obesity (e.g., insulin resistance or different metabolic disorders) [67]. TFAP2B increases the risk of abdominal obesity [68]. As shown in Table 6, the significant differences in the main parameters related to obesity depend on SNPs. 

At the end of Ph2, during the last visit to the Dietitian Service, all participants were asked to complete a survey questionnaire. The questions were focused on taking part in the study and the impact of the COVID-19 pandemic on the weight loss process and eating habits changes. In general, both groups were satisfied with their participation in the study, which allowed them to gain practical knowledge that they could use in the future. In most cases, the COVID-19 lockdown in Poland, which started on 16 March 2020, has an adverse effect on the weight loss process. This situation occurred in Ph2 of our research. The patients declared that they had lost control of their body mass and motivation to reduce it. More details can be found in the Appendix A.

## 4. Discussion

The objective of this study was to assess whether comprehensive post-therapeutic care can help in the long-term maintenance of effects achieved during the bodyweight reduction process, i.e., upgraded body composition and blood analysis results. To the best of our knowledge, this is the first study to evaluate the effectiveness of long-term support on anthropometric and biochemical parameters combined with genetic tests. This study was also unique due to the application of the TTT strategy in combination with some elements of integrating CBT and MI, in Ph2. Habit-based interventions are quite well-known in the weight loss process [69], but only few research teams have undertaken the TTT method in the weight maintenance stage [43].

Our results confirm that the weight loss process is the most effective at the beginning. After this stage, the achieved results begin to change adversely, despite the application of an energy-restricted diet. It is well-known that during caloric restriction, the first phase of weight loss is rapid and lasts less than a week [70,71]. In general, effective weight loss involves greater loss of FM than FFM, especially muscle mass. The second phase is associated with slower weight loss than before, leading to decreased motivation in some participants [72]. This aspect can explain why participants achieved better results after 6 months than 12 months of Ph1. Our results are consistent with a meta-analysis carried out by Johnston et al., who showed that the average weight loss is 6.78 kg after 6 months of a balanced diet (7.57 kg in our study) and 5.70 kg (4.93 kg in our study) after 12 months [73].

Another aspect that may have impeded the weight loss process and weight maintenance in Ph2 was the COVID-19 pandemic. Since the announcement of the COVID-19 pandemic in March 2020, the rapid global spread of the disease has led to many changes in people’s lifestyles and eating habits. Some reports showed an increase in wheat products, sugary food, and snack consumption during self-quarantine, with a simultaneous decrease in the physical activity level [74,75,76]. The results presented above are in line with the findings of this study (Appendix A). The state of lockdown may have led to negative eating patterns and frequent snacking, both associated with higher caloric intake and increased risk of weight regain [77,78], which can explain the present results. Regardless of the difficulties related to the COVID-19 pandemic, the SG group significantly improved anthropometric parameters, considering the total duration of the study (Table 4).

Strategies aimed at reducing bodyweight generally produce favorable short-term results but maintenance of such weight loss often proves difficult in the long term. According to the definition proposed by Wing and Hill, successful maintenance of weight loss consists in “intentional losing ≥10% of initial weight and keeping it off for ≥1 year” [9]. Based on this definition, approximately 80% of obese individuals report unsuccessful maintenance of weight loss and return to the previous bodyweight [Greenway 2015]. Some reports have shown that reducing and maintaining bodyweight by 3–5% brings a significant improvement in lipid and carbohydrate metabolism [79,80]. During phase 2, the participants did not receive diet plans, but only practical tips related to following a healthy energy-restricted diet. Physical activity was also not monitored but the participants were encouraged to maintain its proper level, which, combined with the energy deficit, promotes a more favorable body recomposition than a reduction diet alone [81]. Previous systematic reviews have demonstrated that, after a weight loss program based on diet and exercise, an average of 46–50% of weight loss is regained just one year post-treatment [82,83,84]. However, unlike strict diet and exercise programs, interventions after weight loss focused on permanent habit change encourage the behavior to become ‘a new nature’. The new healthy behavior puts no pressure on the participants; therefore, they are more resistant to change and can last long in time [43], which is confirmed by the present results. Overall, it appears that this phase should be highly comprehensive for weight maintenance. Investigations reporting successful results showed that a combination of energy and fat restriction, regular physical activity, and behavioral strategies are required for long-term weight maintenance [29]. Importantly, these behaviors cannot be imposed on patients but should become their new habit.

Some reports differ from each other in the length of the support intervention. When the weight maintenance phase lasted 18 months, different results were reported in randomized controlled trials. West et al. proved that obese participants that were randomized to motivation-focused or skill-based groups achieved comparable 18-month weight losses (−5.48% for motivation-focused vs. −5.55% in skill-based), and both groups lost significantly more weight than controls (−1.51%) [85]. Jeffrey et al. showed the effectiveness of the maintenance-tailored therapy (MTT) compared to the standard behavior therapy (SBT) for the lengthy treatment of obesity. During 18 months, the mean (SD) weight losses were 8.3 (8.9) kg for MTT and 9.3 (8.8) kg for SBT [86]. These investigations were more effective than our experiment (weight maintenance in SG and a significant increase in CG), and one possible explanation for this is the COVID-19 pandemic, which may have highly impeded the effects obtained in Ph2. The 30-month intervention was also carried out by Yatsuya et.al. [87]. This research was based on CBT and MTT methods. At the end of the study, they showed 1.4 and 2.1 kg/m^2^ decrease in BMI in the group of women and men, respectively. The waist circumference of women and men decreased by 3.4 cm and 5.9 cm, respectively. Our results were more effective in the case of BMI because the control group reduced their BMI by 3.83 kg/m^2^ and similar in waist circumference by 4.83 cm. Different results were obtained for TG and HDL. In our study, the level of TG for 30 months did not change significantly, while the level of HDL decreased. Yatsuya et al. also recorded an initial significant decrease in HDL which increase to values higher than the initial. In turn, the level of TG was dependent on gender. In women, after 30 months, it was significantly higher compared to baseline. In men, the observed results were opposite. Analyzing biochemical results, Wolfson et al. [32] after 30 months of observation showed no significant changes in TG, a significant decrease in total-chol, LDL, and an increase in HDL in the supported group. There were no significant changes in these parameters in the control group. BMI in the supported group decreased by 3.2 kg/m^2^, and in the control group by 0.9 kg/m^2^. A long follow-up period (28 ± 13 months) was also characterized a study conducted by McLaughlin et al. [88]. They proved that the maintenance stage after a 16-week balanced reduction diet can be effective without contact with a specialist. Contrary to our study, after re-contact, the study group significantly decreased their bodyweight (−5.4 kg), BMI (−1.9 kg/m^2^), TG (−41 mg/dL) and increased HDL (+4 mg/dL) levels.

Some research focuses on different times of post-therapeutic care, e.g., 6, 12, and 24 months or 2–4 years. Still, in the studies that met the criteria for successful weight maintenance, there was no relationship between the length of the maintenance phase and the weight maintenance (i.e., the extended care did not result in higher or lower weight regain) [29].

A wide range of anthropometric and biochemical parameters were assessed in this research. This is the first study to analyze such a large number of indicators in the context of weight loss maintenance. Previous studies only assessed changes in the fundamental anthropometric indicators [34,35,89], in a few selected parameters of blood [90], or in both blood and anthropometric parameters, but their number was lower than in the present study [91]. During 30 months, important parameters WC and VAT values in the SG group decreased significantly. In turn, the VAT level in CG increased, and the other parameters remained unchanged. These results confirm the effectiveness of long-term support in improving important metabolic parameters in obese individuals. Among the blood biochemical parameters that were not discussed in the “Results” Section, CRP level seems to be important in the context of weight loss. There is strong evidence of effective weight loss as a nonpharmacologic strategy for lowering CRP levels [92], especially in obese patients characterized by increased circulating levels of proinflammatory cytokines [93]. The present results confirm these theses. As a result of the energy-restricted diet, CRP decreased significantly (*p* < 0.05) in the whole group during the first 12 months, while a significant increase in CRP in CG and no change in SG were observed in the maintenance phase.

The IPAQ questionnaire is commonly used in research practice to assess physical activity levels. Using the search term “IPAQ questionnaire” in PubMed, it is clear that there has been a systematically increasing annual number of publications focused on this topic, with only one in 2001 and 219 articles in 2020. For obese individuals, varying levels of total physical activity have been reported. Average values 756.51 ± 334.95 MET-min/week were reported for older people [94]. Middle-aged individuals showed higher values: 1214.6 ± 1171.6 MET-min/week for men and 1046.8 ± 1133.2 MET-min/week for women [95]. Overweight and obese youths are characterized by a considerably higher level of physical activity, i.e., 2761.0 MET-min/week [96]. Our results showed a significant increase in the level of physical activity during Ph1. This fact was probably associated with improvement of the effects of the weight loss process. During the maintenance phase, the physical activity levels decreased in both groups (*p* < 0.05 in CG), which may have been related to the COVID-19 lockdown. While self-quarantine is a necessary measure to protect public health, the results indicate that it alters physical activity and eating behaviors in a health-compromising direction [97].

This research also investigated the genetic predisposition to obesity in the studied group and the impact of the tested SNPs on reduced bodyweight and maintenance of the achieved effects (Table 6). In their meta-analyses, Doaei et al. have reported that FTO rs9930506 polymorphisms play a crucial role in regulating bodyweight and BMI values and are strongly related to obesity [50]. Some data suggest that subjects with the rs9939609 risk allele may have more difficulty maintaining fat mass after weight loss intervention than adults and children with other genotypes [98,99]. Our results are compatible with the data mentioned above. Individuals with the A allele (AA or AT) were able to lose weight, fat mass, BMI, and VAT, but their results were significantly (*p* < 0.05) less effective than those of subjects with TT variants. Moreover, the AA genotype showed higher values of anthropometric parameters than the baseline at the beginning of the study. This relationship was not observed in individuals with the T allele. We also investigated the association of PLIN1 rs894160 polymorphisms with body composition changes and their long-term maintenance. The present study shows a possible effect of the PLIN1 (rs894160) polymorphism in modulating anthropometric changes induced by an energy-restricted diet intervention. Obese individuals with the A allele had a higher decrease in bodyweight and total FM after the 12-month energy-restricted diet intervention. Additional allele A predisposes to more effective maintenance of achieved effects than the G allele. These results are in agreement with Soenen et al., who showed larger weight loss and loss of fat mass in women with the PLIN1 (rs894160) A-alleles. They reported a differential response to weight loss-weight maintenance intervention supported by a differential effect on body composition, with the role of the PLIN genotype therein [100]. On the other hand, reduced diet resulted in significant decreases in bodyweight in GG individuals. Individuals with the A allele may have greater difficulties in reducing bodyweight or central body fat [101]. Similar conclusions were presented by Ruiz et al., who reported that a reduced lipid oxidation rate might partially explain this phenomenon it. They observed a lower lipid oxidation rate in individuals with the A allele, and it was significantly associated with changes in FM measured by dual-energy X-Ray absorptiometry. This fact is interesting, since it is known that individuals with lower lipid oxidation rates after weight loss are most susceptible to weight regain [67,102]. Due to the relatively small sample size in the present study (especially the AA variant, n = 2), these findings should be regarded as preliminary. TFAP2B rs987237 is associated with obesity and has shown interaction with the dietary fat-to-carbohydrate ratio, affecting weight loss [103,104]. In most studies, the TFAP2B (GG) genotype was negatively associated with obesity. Contrary to our results, Stocks et al. reported that the AA genotype was beneficial for weight maintenance. Still, in the AG and GG groups, no differences were observed after diet with different protein content. On a high-protein diet, carriers of the obesity risk allele (G allele) regained 1.84 kg more bodyweight per risk allele than individuals on a low-protein diet after a 6-month weight maintenance period [105]. Our results show a significant bodyweight regain only in the AG and GG individuals. This can be explained by the small sample size or the application of the same balanced diet for all participants.

This investigation has several strengths, including data from methodologically rigorous randomized controlled trials, e.g., the reliable measure pair-matching method, an intuitive design strategy to protect study validity and increase study power [106]. Additionally, the monitoring system was also well managed. It was long-term lifestyle-based research. The post-therapeutic support period in the present study was substantially longer than in previous studies in this field. Ph2 compared to Ph1 was half a time longer, which allowed better assessment of the results obtained at this stage. Another strength of this study is the fact that this investigation was designed and/or conducted by an interdisciplinary team consisting of a dietitian (K.I.), nutritionist (P.G.), doctor of medicine (M.S.), and psychologist (W.S.). This aspect represents a multidimensional approach to the problem and allows implementation of many practical elements into the therapy.

The limitations of this study include the small number of participants, which, on the other hand, facilitated an individual and personal approach to every patient. The number of participants considered in our study is similar to that reported by some previous studies investigating weight loss maintenance following dietary treatment of obesity [107,108,109,110,111,112]. This fact needs to be considered when attempting to apply the findings of the present study to the broader population as a potential public health intervention to manage obesity. We did not use a strict control of consumption during Ph1, and of physical activity level during all study, which may also have influenced the obtained results. Another weakness of our study was the exclusion of those who had metabolic and autoimmune disorders, which made our results difficult to generalize to the average patients seeking medical assistance with weight loss.

## 5. Conclusions

In conclusion, the present results demonstrate that long-term dietitian and psychological support of obese patients who reduced their weight allows them to avoid weight regain. This study supports healthy nutritional habits as a behavioral tool with the potential to encourage long-term weight loss and maintenance. On the other hand, the interplay of genetic obesity risk factors is not fully understood, and our results suggest that these factors may play some role in weight regain. Future investigations should clarify whether SNPs associated with obesity in cross-sectional settings overlap with those contributing to bodyweight changes in clinical weight loss interventions. Post-therapeutic support to current clinical practice may increase the chances of long-term weight loss maintenance. However, future long-term research on a larger number of people is needed to confirm whether comprehensive support makes weight loss and weight maintenance less challenging and more successful.

## Figures and Tables

**Figure 1 nutrients-13-02020-f001:**
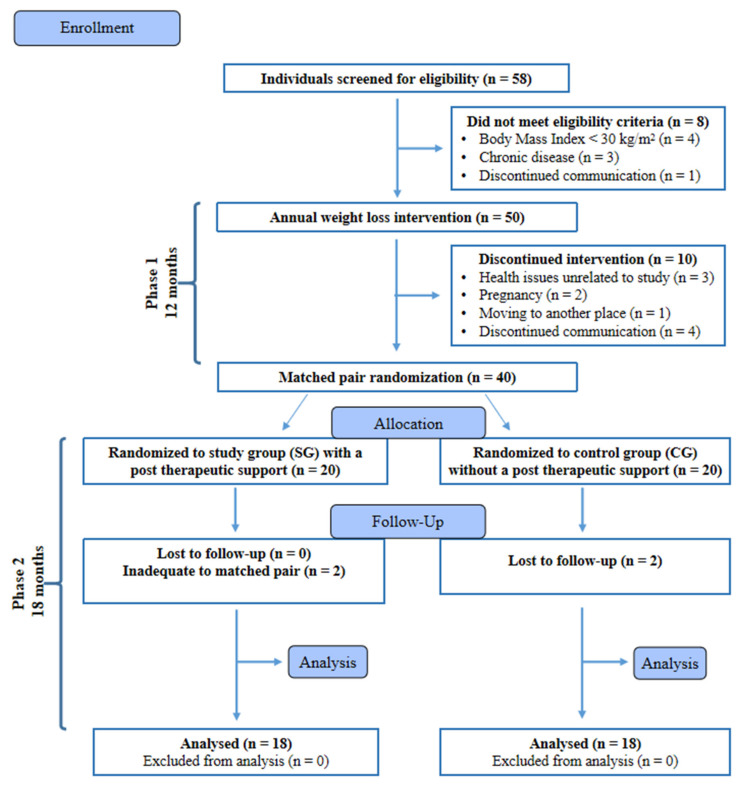
Flow diagram of the patient recruitment and randomization process. Adapted from: CONSORT 2010 Flow Diagram [49].

**Figure 2 nutrients-13-02020-f002:**
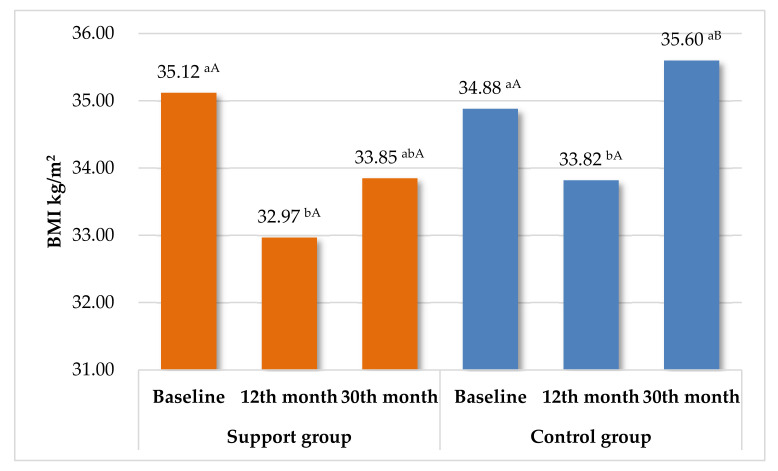
Summary changes in BMI during the entire study in SG and CG. Means with different letters are significantly different at *p* < 0.05; ^ab^—differences between the changes in BMI within each group separately (ANOVA with repeated measures—Tukey test). ^AB^—significant differences between SG and CG in each checkpoint (Student’s *t*-test).

**Table 1 nutrients-13-02020-t001:** Characteristics of the studied group during Ph1 (weight loss phase) of the intervention.

Parameter	Baseline (n = 36)	After a 6 Months of Ph1 (n = 36)	After a 12 Months of Ph1 (n = 36)
	Women (n = 22), Men (n = 14)	Women (n = 22), Men (n = 14)	Women (n = 22), Men (n = 14)
Bodyweight (kg)	104.0 ^A^ ± 17.92	96.63 ^C^ ± 18.25	99.27 ^B^ ± 18.86
BMI (kg/m^2^)	35.04 ^A^ ± 3.80	32.43 ^C^ ± 3.96	33.40 ^B^ ± 4.12
Fat mass (kg)	42.55 ^A^ ± 9.03	36.77 ^C^ ± 9.53	39.40 ^B^ ± 10.46
Fat mass (%)	41.11 ^A^ ± 5.96	38.23 ^C^ ± 6.64	39.40 ^B^ ± 10.46
VAT (L)	3.69 ^A^ ± 2.36	2.65 ^B^ ± 1.99	2.82 ^B^ ± 1.99
FMI value (kg/m^2^)	14.46 ^A^ ± 3.03	12.47 ^C^ ± 3.13	13.35 ^B^ ± 3.37
WC (cm)	107.72 ^A^ ± 13.56	99.42 ^C^ ± 13.50	101.17 ^B^ ± 13.34
Free fat mass (kg)	61.58 ^A^ ± 13.30	59.77 ^B^ ± 13.45	59.87 ^B^ ± 12.86
Muscle mass (kg)	30.32 ^A^ ± 7.52	28.99 ^B^ ± 7.66	29.07 ^B^ ± 7.28
Total body water (L)	45.61 ^A^ ± 9.50	44.01 ^B^ ± 9.72	44.35 ^B^ ± 9.28
Phase angle (°)	5.57 ^A^ ± 0.65	5.54 ^A^ ± 0.68	5.55 ^A^ ± 0.65
Total IPAQ value (MET-min/week)	1742.34 ^B^ ± 1746.80	-	1993.78 ^A^ ± 1543.90

^ABC^ Means with different letters in the same row are significantly different at *p* < 0.05 (ANOVA with repeated measures—Tukey test). Abbreviations: BMI (Body Mass Index), VAT (Visceral Adipose Tissue), FMI (Fat Mass Index), WC (Waist Circumference), IPAQ (International Physical Activity Questionnaire).

**Table 2 nutrients-13-02020-t002:** Biochemical parameters changes in all subjects during Ph1 (weight loss phase).

Parameter	Baseline (n = 36)	After a 12 Months of Ph1 (n = 36)	Reference Range		*p*-Value	Effect Size (Cohen’s d)
	Women (n = 22), Men (n = 14)	Women (n = 22), Men (n = 14)	Men	Women		
Glucose (mg/dL)	98.36 ± 8.7	97.83 ± 9.61	70.0–99.0		0.69021	0.06
Total-chol (mg/dL)	190.39 ± 36.09	204.22 ± 38.60	115.0–190.0		0.02131 *	0.37
HDL (mg/dL)	53.00 ± 14.62	54.50 ± 14.08	>40.0	>45.0	0.42221	0.10
LDL (mg/dL)	115.08 ± 30.92	127.47 ± 32.14	Annotation ^1^		0.02752 *	0.39
TG (mg/dL)	114.61 ± 50.36	112.02 ± 39.65	35.0–150.0		0.72768	0.06
AST (U/L)	29.64 ± 25.96	20.25 ± 6.08	<40.0	<32.0	0.00081 *	0.50
ALT (U/L)	38.20 ± 36.67	25.12 ± 17.89	<41.0	<33.0	0.00086 *	0.45
CRP (mg/L)	2.68 ± 2.51	1.96 ± 2.11	<5.0		0.02784 *	0.31
TSH (mIU/L)	2.14 ± 1.08	1.99 ± 0.74	0.27–4.20		0.52973	0.16
Leukocytes (× 10^3^/µL)	6.24 ± 1.40	5.88 ± 1.10	4.0–10.0		0.06375	0.29
Erythrocytes (× 10^6^/dL)	4.98 ± 0.41	4.89 ± 0.41	4.5–6.0	3.8–5.4	0.01576 *	0.22
Hemoglobin (g/dL)	14.45 ± 1.66	14.28 ± 1.45	14.0–18.0	12.0–16.0	0.10349	0.11
Platelets (× 10^3^/µL)	263.25 ± 71.28	256.11 ± 66.17	130.0–400.0		0.41846	0.10

* *p*-value—statistically significant difference between a group at baseline and 12 months at *p* < 0.05 (Student’s *t*-test). Effect size (Cohen’s d)—suggest that d ≈ 0.2 be considered a small effect size, d ≈ 0.5 represents a medium effect size and d ≈ 0.8 a large effect size. Annotation ^1^—based on cardiological prevention guidelines, the range of the LDL norm depends on the level of cardiovascular risk and cardiac comorbidities [57]. Abbreviations: HDL (high-density lipoprotein), LDL (low-density lipoprotein), TG (triglycerides), AST (aspartate transaminase), ALT (alanine aminotransferase), CRP (C-reactive protein), TSH (thyroid stimulating hormone).

**Table 3 nutrients-13-02020-t003:** Baseline, month 18, and changes in anthropometric and biochemical parameters within and between supported (SG) and no additional care (CG) in Ph2.

Parameter	SG (n = 18)			*p*-Value	CG = (n = 18) Women (n = 11), Men (n = 7)			*p*-Value
	Women (n = 11), Men (n = 7)							
	Before Ph2	After 18 Months	Change		Before Ph2	After 18 Months	Change	
Bodyweight (kg)	97.52 ± 14.79	100.04 ± 15.62	2.52 ± 8.4	0.21966	101.02 ± 22.51	106.33 ± 23.90	5.31 ± 4.62	0.00014 *
BMI (kg/m^2^)	32.97 ± 3.29	33.85 ± 3.70	0.88 ± 2.85	0.20617	33.82 ± 4.88	35.60 ± 5.23	1.78 ± 1.41	0.00055 *
Fat mass (kg)	37.74 ± 8.88	40.28 ± 9.93	2.54 ± 7.96	0.19334	41.06 ± 11.85	44.84 ± 11.90	3.78 ± 3.29	0.00014 *
Fat mass (%)	38.74 ± 6.89	40.24 ± 6.45	1.51 ± 4.80	0.20095	40.45 ± 6.38	42.15 ± 5.80	1.69 ± 2.30	0.00611 *
VAT (L)	2.75 ± 1.87	3.20 ± 2.47	0.45 ± 1.24	0.14456	2.90 ± 2.16	4.04 ± 2.87	1.14 ± 1.08	0.00033 *
FMI value (kg/m^2^)	12.86 ± 3.05	13.72 ± 3.02	0.86 ± 2.70	0.19308	13.84 ± 3.69	15.15 ± 3.75	1.31 ± 1.05	0.00006 *
WC (cm)	100.7 ± 11.5	102.94 ± 13.81	2.3 ± 7.0	0.18790	101.67 ± 15.30	109.50 ± 17.77	7.83 ± 5.97	0.00003 *
Free fat mass (kg)	59.78 ± 11.49	59.76 ± 11.18	−0.02 ± 1.31	0.67907	59.96 ± 14.44	61.49 ± 15.62	1.53 ± 2.91	0.03961 *
Muscle mass (kg)	28.93 ± 6.63	28.98 ± 6.40	0.05 ± 1.33	0.77715	29.21 ± 8.06	30.09 ± 8.91	0.87 ± 1.79	0.05366
Total body water (L)	44.13 ± 8.20	44.21 ± 7.87	0.08 ± 1.31	0.87894	44.57 ± 10.48	45.70 ± 11.48	1.13 ± 2.33	0.05515
Phase angle (°)	5.66 ± 0.67	5.61 ± 0.70	−0.05 ± 0.19	0.71742	5.44 ± 0.62	5.43 ± 0.60	−0.01 ± 0.21	0.30391
IPAQ value (MET-min/week)	1901.28 ± 932.66	1700.78 ± 1108.10	−200.50 ± 876.08	0.34517	2086.28 ± 2004.86	1636.83 ± 2000.16	−449.44 ± 717.06	0.00433 *
Biochemical parameters								
Glucose (mg/dL)	96.50 ± 5.40	101.83 ± 11.49	5.33 ± 9.55	0.02990 *	99.17 ± 12.53	108.06 ± 13.26	8.89 ± 6.26	0.00001 *
Total-chol (mg/dL)	207.17 ± 38.80	201.39 ± 25.60	−5.78 ± 30.92	0.43877	201.28 ± 39.29	204.61 ± 42.63	3.33 ± 22.04	0.52965
HDL (mg/dL)	57.33 ± 16.04	52.22 ± 15.03	−5.11 ± 5.89	0.00185 *	51.67 ± 11.57	49.33 ± 9.41	−2.33 ± 5.12	0.05911
LDL (mg/dL)	128.33 ± 30.70	124.72 ± 22.20	−3.61 ± 27.38	0.58305	126.61 ± 34.39	130.00 ± 36.21	3.39 ± 24.53	0.56554
TG (mg/dL)	108.29 ± 36.93	122.42 ± 51.24	14.13 ± 37.77	0.13093	115.75 ± 42.93	133.46 ± 52.29	17.71 ± 41.14	0.08550
AST (U/L)	20.49 ± 6.51	21.88 ± 9.28	1.38 ± 5.70	0.52772	20.01 ± 5.78	22.41 ± 8.58	2.40 ± 6.03	0.19889
ALT (U/L)	26.28 ± 21.43	29.54 ± 26.30	3.26 ± 11.95	0.77711	23.96 ± 14.02	29.96 ± 18.85	5.99 ± 11.77	0.01474 *
CRP (mg/L)	1.54 ± 1.14	2.19 ± 1.66	0.65 ± 1.58	0.09896	2.39 ± 2.74	3.26 ± 3.81	0.88 ± 1.75	0.03467 *
TSH (mIU/L)	2.08 ± 0.82	1.90 ± 0.92	−0.18 ± 0.84	0.38051	1.90 ± 0.64	1.96 ± 0.60	0.06 ± 0.67	0.72755
Leukocytes (× 10^3^/µL)	5.92 ± 1.05	6.19 ± 1.23	0.27 ± 1.01	0.27544	5.83 ± 1.18	6.52 ± 1.44	0.70 ± 1.13	0.01660 *
Erythrocytes (× 10^6^/dL)	4.85 ± 0.44	4.84 ± 0.42	−0.01 ± 0.28	0.89439	4.93 ± 0.37	4.96 ± 0.33	0.03 ± 0.18	0.46419
Hemoglobin (g/dL)	14.32 ± 1.55	14.14 ± 1.57	−0.18 ± 0.79	0.26677	14.24 ± 1.38	14.43 ± 1.15	0.19 ± 0.67	0.24707
Platelets (× 10^3^/µL)	238.22 ± 37.23	245.39 ± 45.19	7.17 ± 40.13	0.45903	274.00 ± 83.37	273.56 ± 81.91	−0.44 ± 19.51	0.92414

* *p*-value—statistically significant difference between “Before Ph2” and “After 18 months” at *p* < 0.05 (Student’s *t*-test). Abbreviations: SG (support group), CG (control group), BMI (Body Mass Index), VAT (Visceral Adipose Tissue), FMI (Fat Mass Index), WC (Waist Circumference), IPAQ (International Physical Activity Questionnaire), HDL (high-density lipoprotein), LDL (low-density lipoprotein), TG (triglycerides), AST (aspartate transaminase), ALT (alanine aminotransferase), CRP (C-reactive protein), TSH (thyroid stimulating hormone).

**Table 4 nutrients-13-02020-t004:** Changes in anthropometric and biochemical parameters within and between the SG and CG groups throughout the study.

Parameter	SG (n = 18)			*p*-Value	CG (n = 18)			*p*-Value
	Women (n = 11), Men (n = 7)				Women (n = 11), Men (n = 7)			
	Before Ph1	After 30 Months	Change		Before Ph1	After 30 Months	Change	
Bodyweight (kg)	103.86 ± 14.98	100.04 ± 15.62	−3.83 ± 6.09	0.01634 *	103.84 ± 20.66	106.33 ± 23.90	2.48 ± 6.24	0.10954
BMI (kg/m^2^)	35.12 ± 3.24	33.85 ± 3.70	−1.27 ± 2.02	0.01598 *	34.88 ± 4.29	35.60 ± 5.23	0.72 ± 2.12	0.16987
Fat mass (kg)	42.31 ± 8.03	40.28 ± 8.93	−2.03 ± 5.52	0.13663	42.79 ± 10.15	44.84 ± 11.90	2.05 ± 4.99	0.09994
Fat mass (%)	40.86 ± 5.82	40.24 ± 6.45	−0.62 ± 3.43	0.45566	41.36 ± 6.26	42.15 ± 5.80	0.78 ± 2.64	0.22486
VAT (L)	3.79 ± 2.47	3.20 ± 2.47	−0.58 ± 0.63	0.00378 *	3.59 ± 2.32	4.04 ± 2.87	0.45 ± 0.74	0.01958 *
FMI value (kg/m^2^)	14.41 ± 2.78	13.72 ± 3.02	−0.69 ± 1.82	0.12622	14.51 ± 3.35	15.15 ± 3.75	0.64 ± 1.70	0.12852
WC (cm)	107.78 ± 13.33	102.94 ± 13.81	−4.83 ± 4.05	0.00009 *	107.67 ± 14.18	109.50 ± 17.77	1.83 ± 5.97	0.21021
Free fat mass (kg)	61.56 ± 11.62	59.76 ± 11.18	−1.79 ± 2.42	0.00956 *	61.05 ± 14.98	61.49 ± 15.62	0.43 ± 2.54	0.47976
Muscle mass (kg)	30.57 ± 6.61	28.98 ± 6.40	−1.59 ± 1.51	0.00100 *	29.81 ± 8.38	30.09 ± 8.91	0.27 ± 1.34	0.40120
Total body water (L)	45.92 ± 8.23	44.21 ± 7.87	−1.70 ± 1.96	0.00285 *	45.30 ± 10.85	45.70 ± 11.48	0.40 ± 2.03	0.41932
Phase angle (°)	5.67 ± 0.64	5.61 ± 0.70	−0.06 ± 0.28	0.38562	5.48 ± 0.66	5.43 ± 0.60	−0.04 ± 0.23	0.75831
IPAQ value (MET-min/week)	1663.72 ± 1269.16	1700.78 ± 1108.10	37.06 ± 1064.92	0.88435	1820.97 ± 2158.29	1636.83 ± 2000.16	−184.14 ± 661.15	0.55657
Biochemical parameters								
Glucose (mg/dL)	99.22 ± 7.37	101.83 ± 11.49	2.61 ± 10.99	0.32793	97.50 ± 9.99	108.06 ± 13.26	10.56 ± 8.51	0.00006 *
Total-chol (mg/dL)	197.78 ± 26.65	201.39 ± 25.60	3.61 ± 29.17	0.60620	183.00 ± 43.08	204.61 ± 42.63	21.61 ± 7.62	0.00405 *
HDL (mg/dL)	56.78 ± 16.68	52.22 ± 15.03	−4.56 ± 5.87	0.00432 *	49.22 ± 1.46	49.33 ± 9.41	0.11 ± 5.97	0.93798
LDL (mg/dL)	118.06 ± 28.14	124.72 ± 22.20	6.67 ± 29.49	0.35091	112.11 ± 34.03	130.00 ± 36.21	17.89 ± 20.99	0.00214
TG (mg/dL)	121.09 ± 52.98	122.42 ± 51.24	1.33 ± 57.16	0.92233	108.13 ± 48.22	133.46 ± 52.29	25.32 ± 49.43	0.05263
AST (U/L)	25.53 ± 13.90	21.88 ± 9.28	−3.66 ± 8.94	0.14458	33.74 ± 34.04	22.41 ± 8.58	−11.33 ± 32.41	0.08539
ALT (U/L)	43.09 ± 47.59	29.54 ± 26.30	−13.54±25.45	0.01226 *	33.31 ± 21.30	29.96 ± 18.85	−3.36 ± 20.18	0.49965
CRP (mg/L)	2.99 ± 2.66	2.19 ± 1.66	−0.80 ± 2.90	0.24847	2.36 ± 2.38	3.26 ± 3.81	−0.90 ± 1.74	0.02491 *
TSH (mIU/L)	2.19 ± 1.11	1.90 ± 0.92	−0.30 ± 1.11	0.26975	2.08 ± 1.08	1.96 ± 0.60	−0.12 ± 0.99	0.74395
Leukocytes (× 10^3^/µL)	6.06 ± 1.24	6.19 ± 1.23	0.13 ± 1.25	0.67683	6.41 ± 1.55	6.52 ± 1.44	0.11 ± 1.37	0.32017
Erythrocytes (× 10^6^/dL)	4,92 ± 0.50	4.84 ± 0.42	−0.08 ± 0.27	0.23162	5.03 ± 0.30	4.96 ± 0.33	−0.67 ± 0.17	0.11306
Hemoglobin (g/dL)	14.44 ± 1.95	14.14 ± 1.57	0.31 ± 1.17	0.28467	14.46 ± 1.37	14.43 ± 1.15	−0.03 ± 0.60	0.81619
Platelets (× 10^3^/µL)	247.94 ± 48.57	245.39 ± 45.19	−2.56 ± 53.63	0.84218	278.56 ± 87.22	273.56 ± 81.91	−5.00 ± 48.12	0.66490

* *p*-value—statistically significant difference between “Before Ph1” and “After 30 months” at *p* < 0.05 (Student’s *t*-test). Abbreviations: BMI (Body Mass Index), VAT (Visceral Adipose Tissue), FMI (Fat Mass Index), WC (Waist Circumference), IPAQ (International Physical Activity Questionnaire), HDL (high-density lipoprotein), LDL (low-density lipo-protein), TG (triglycerides), AST (aspartate transaminase), ALT (alanine aminotransferase), CRP (C-reactive protein), TSH (thyroid stimulating hormone).

**Table 5 nutrients-13-02020-t005:** SNPs in the study group.

Parameter	Baseline (n = 36)	SG (n = 18)	CG (n = 18)
rs9939609 (FTO)			
AA	10 (28%)	4	6
AT	22 (61%)	12	10
TT	4 (11%)	2	2
rs987237 (TFAP2B)			
GG	4 (11%)	1	3
AG	16 (44%)	7	9
AA	16 (44%)	10	6
rs894160 (PLIN1)			
AA	2 (6%)	1	1
AG	15 (42%)	9	6
GG	19 (53%)	8	11

**Table 6 nutrients-13-02020-t006:** Differences in changes in the anthropometric parameters at the best point of the study (6 month) compared with 30 month, depending on analyzed single nucleotide polymorphisms (SNPs).

Parameter	rs9939609 (FTO)						rs987237 (TFAP2B)						rs894160 (PLIN1)					
	AA (n = 10)		AT (n = 22)		TT (n = 4)		AA (n = 16)		AG (n = 16)		GG (n = 4)		AA (n = 2)		AG (n = 15)		GG (n = 19)	
	6 Month	30 Month	6 Month	30 Month	6 Month	30 Month	6 Month	30 Month	6 Month	30 Month	6 Month	30 Month	6 Month	30 Month	6 Month	30 Month	6 Month	30 Month
Bodyweight (kg)	−5.71 ^bB^ ± 4.09	2.89 ^aA^ ± 7.69	−8.62 ^bB^ ± 4.87	−1.89 ^aAB^ ± 6.44	−6.89 ^aB^ ± 2.58	−2.84 ^aAB^ ± 4.84	−8.07 ^bA^ ± 5.62	0.10 ^aA^ ± 6.54	−7.04 ^bA^ ± 3.97	−2.10 ^aA^ ± 6.63	−8.15 ^bA^ ± 1.71	0.44 ^aA^ ± 11.26	−11.20 ^aB^ ± 2.83	−9.95 ^aB^ ± 2.26	−7.64 ^bAB^ ± 2.60	−1.86 ^aAB^ ± 7.86	−7.23 ^bAB^ ± 5.76	1.24 ^aA^ ± 5.37
BMI (kg/m^2^)	−1.95 ^bB^ ± 1.47	1.03 ^aA^ ± 2.39	−3.01 ^bB^ ± 1.67	−0.73 ^aAB^ ± 2.16	−2.26 ^aB^ ± 0.83	−1.05 ^aAB^ ± 1.59	−2.78 ^bB^ ± 1.95	0.18 ^aA^ ± 2.06	−2.50 ^bAB^ ± 1.40	−0.80 ^aAB^ ± 2.19	−2.60 ^aAB^ ± 0.60	0.12 ^aA^ ± 3.51	−4.09 ^aC^ ± 1.00	−3.54 ^aBC^ ± 0.92	−2.66 ^bABC^ ± 0.98	−0.64 ^aAB^ ± 2.57	−2.46 ^bABC^ ± 1.96	0.35 ^aA^ ± 1.78
Fat mass (kg)	−4.18 ^bB^ ± 3.08	3.14 ^aA^ ± 6.12	−6.44 ^bB^ ± 4.85	−0.89 ^aAB^ ± 5.34	−6.10 ^aB^ ± 2.53	−2.86 ^aB^ ± 1.67	−6.45 ^bBB^ ± 4.73	1.27 ^aA^ ± 5.10	−5.17 ^bAB^ ± 4.33	−1.35 ^aAB^ ± 5.09	−5.50 ^aAB^ ± 1.27	1.09 ^aA^ ± 9.24	−7.92 ^aA^ ± 2.56	−6.55 ^aA^ ± 1.77	−5.41 ^aA^ ± 2.67	−0.78 ^aA^ ± 6.13	−5.83 ^bA^ ± 5.36	1.32 ^aA^ ± 4.92
Fat mass (%)	−1.96 ^bABC^ ± 1.85	1.30 ^aA^ ± 3.55	−3.21 ^bBC^ ± 3.47	−0.18 ^aAB^ ± 3.05	−3.42 ^aBC^ ± 1.48	−1.49 ^aBC^ ± 0.64	−3.38 ^bB^ ± 2.83	1.25 ^aA^ ± 3.13	−2.61 ^aAB^ ± 3.38	−0.82 ^aAB^ ± 3.09	−2.02 ^aAB^ ± 0.92	0.42 ^aAB^ ± 3.93	−4.56 ^aA^ ± 1.33	−2.88 ^aA^ ± 1.24	−2.70 ^aA^ ± 2.60	−0.39 ^aA^ ± 3.55	−2.86 ^bA^ ± 3.33	0.76 ^aA^ ± 2.67
VAT (L)	−0.82 ^bAB^ ± 0.64	0.32 ^aA^ ± 1.12	−1.11 ^bB^ ± 1.00	−0.25 ^aAB^ ± 0.69	−1.13 ^aB^ ± 0.64	0.00 ^aAB^ ± 0.83	−1.20 ^bB^ ± 0.98	−0.14 ^aAB^ ± 0.81	−0.89 ^bB^ ± 0.77	−0.14 ^aAB^ ± 0.77	−0.93 ^aB^ ± 0.95	0.53 ^aA^ ± 1.35	−0.73 ^aA^ ± 0.47	−0.43 ^aA^ ± 0.22	−0.89 ^bA^ ± 0.59	−0.15 ^aA^ ± 0.82	−1.17 ^bA^ ± 1.07	0.03 ^aA^ ± 0.93
WC (cm)	−6.30 ^bBC^ ± 4.03	1.30 ^aA^ ± 6.15	−9.41 ^bC^ ± 4.35	−2.91 ^aABC^ ± 5.81	−7.25 ^aBC^ ± 2.50	−0.75 ^aAB^ ± 6.13	−8.75 ^bB^ ± 4.85	−1.50 ^aAB^ ± 5.40	−8.25 ^bB^ ± 3.70	−2.00 ^aAB^ ± 6.42	−6.75 ^aB^ ± 4.57	0.50 ^aA^ ± 8.35	−8.50 ^aA^ ± 0.71	−8.50 ^aA^ ± 2.12	−8.27 ^bA^ ± 3.01	−1.93 ^aA^ ± 6.71	−8.32 ^bA^ ± 5.30	−0.42 ^aA^ ± 5.41

^abABC^—differences between the changes in anthropometric parameters in genes separately. ^ab^—significant differences between 6th and 30th month in each SNP at *p* < 0.05; ^ABC^—significant differences between all SNPs at *p* < 0.05. Abbreviations: BMI (Body Mass Index), VAT (Visceral Adipose Tissue), WC (Waist Circumference).

## Data Availability

The data presented in this study are available on request from the corresponding author. The data are not publicly available due to restrictions concerning privacy.

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
