# Peer review of "The Long-Term Dietitian and Psychological Support of Obese Patients Who Have Reduced Their Weight Allows Them to Maintain the Effects"

_nutrients, 2021, doi:10.3390/nu13062020_

Round 1

Reviewer 1 Report

The manuscript "Only Long-Term Dietitian and Psychological Support of Obese Patients Who Have Reduced Their Weight Allows Them to Maintain the Effects" is a randomized control trial evaluating the long-term effect of diet therapy and behavioral support on metabolic and anthropometric parameters and the association of different obesity genotypes with these changes - among 36 adults. 

I find the methodology of the research appropriate and impressed by the long follow-up time presented. Still, there are several issues raised :

A. Major:

  1. Although a randomized controlled trial is considered the state of the art for the authors' question, an observational study can also be appropriate with larger sample size. Actually, the trial presents the efficacy of a long-term nutritional intervention for 30 months. In the current version, it is not so clear what is the difference between Phase 1 and phase 2 regarding nutritional intervention. I suggest that the authors add a clear diagram presenting the program's visits and "highlights" of the program in its different phases. 
  2. The topic of the paper is very definitive and should be changed. The study is not appropriate to answer if long-term nutritional and behavioral support is the ONLY way to maintain weight loss achievements. Remove the word "only" from the topic as well as from the discussion section. 
  3. The background and the discussion, in particular, are very long and in some parts seem like a narrative review of different topics of obesity. If the paper aims to present data on the maintenance of weight loss achievements, then comprehensive data on weight loss maintenance should be presented in the background ending with the aims of the current study and what was not investigated and is presented in this study. All the information on BMI, waist circumference, etc is known and irrelevant in the background and again in the discussion. Maybe the authors can differentiate their intervention based on the program they relied on - if using the 10 tips + CBT  has not been investigated previously - they can mention that after reviewing previous data. The discussion should be shortened and show comparative data of previous trials with long-term programs supporting weight loss. Then the authors can compare the effects of their program on major metabolic parameters as compared to those shown in other studies. 
  4. Please consider adding some figures showing the efficacy of the intervention in its different phases (compared to the control group from phase 2 until the end of follow up) on BMI, % fat or VAT, and the main metabolic parameters which the authors want to emphasize and discuss (glucose ?...others?). 

Minor:

  1. Statistical methods : change "measurable parameters" to continuous parameters. Also "non-measurable" into categorical/ numerical ...
  2. Tables: a list of abbreviations should be presented at the bottom of the table. Preferably, the statistical method used for comparison within the table should be mentioned. 
  3. Results - please add some written description on tables 5 and 6 

Author Response

Reviewer 1 Changes in the manuscript are highlighted in green (reviewer 2 - changes highlighted in turquoise, editorial office - changes highlighted in yellow)

We would like to thank the Reviewer 1 for the valuable comments and efforts towards improving our manuscript. In the following, we highlight comments and suggestions from Reviewer 1 and our effort to address these remarks.

Point 1: Although a randomized controlled trial is considered the state of the art for the authors' question, an observational study can also be appropriate with larger sample size. Actually, the trial presents the efficacy of a long-term nutritional intervention for 30 months. In the current version, it is not so clear what is the difference between Phase 1 and phase 2 regarding nutritional intervention. I suggest that the authors add a clear diagram presenting the program's visits and "highlights" of the program in its different phases.”

Reply 1: We thank the Reviewer for this remark. However, we believe that the differences in nutritional interventions between Phase 1 and Phase 2 are clearly presented in the text in lines:

148: In this phase (phase 1), special attention was paid to personalization of the nutritional plan and an individual approach to the patient. The details of the intervention have been described in our previous paper [41]. Briefly, A nutrition interview was conducted, and five-day food diaries prepared by the participants were analyzes to compose a personal diet for each patient. The diet plan met all the assumptions (qualitative and quantitative) of a healthy, balanced diet, and included 4 or 5 meals per day.

167: After Ph1, CG did not receive any dietary care. In contrast, SG was invited to participate in group and individual meetings conducted by a qualified dietitian and a psychologist. The main assumptions in this stage comprised regular dietary education, in particular focused on the most practical information of balanced nutrition, and psychological support. Based on the acquired knowledge, the participants were expected to become independent in weight-loss maintenance without applying a strict nutritional plan.

484: “During phase 2, the participants did not receive diet plans, but only practical tips related to following a healthy energy-restricted diet.”

To respond to the Reviewer’s concern, we have included a chart showing the details of the nutritional intervention used in phase 1 and phase 2 in the supplementary materials “FigureS1”.

Point 2: The topic of the paper is very definitive and should be changed. The study is not appropriate to answer if long-term nutritional and behavioral support is the ONLY way to maintain weight loss achievements. Remove the word "only" from the topic as well as from the discussion section”

Reply 2: We agree with the reviewer on this point. We have modified the title to: “The Long-Term Dietitian and Psychological Support of Obese Patients Who Have Reduced Their Weight Allows Them to Maintain the Effects” and we have removed the word "only" from the discussion and conclusion sections.

Point 3: The background and the discussion, in particular, are very long and in some parts seem like a narrative review of different topics of obesity. If the paper aims to present data on the maintenance of weight loss achievements, then comprehensive data on weight loss maintenance should be presented in the background ending with the aims of the current study and what was not investigated and is presented in this study. All the information on BMI, waist circumference, etc is known and irrelevant in the background and again in the discussion. Maybe the authors can differentiate their intervention based on the program they relied on - if using the 10 tips + CBT  has not been investigated previously - they can mention that after reviewing previous data. The discussion should be shortened and show comparative data of previous trials with long-term programs supporting weight loss. Then the authors can compare the effects of their program on major metabolic parameters as compared to those shown in other studies.”

Reply 3: We thank the Reviewer for raising this critical issue. To respond to the Reviewer’s concern, we have removed some general information about obesity from the introduction. We believe that information on BMI and WC, despite being widely known, is inextricably linked to the diagnosis of obesity, so we have not completely deleted it. In the introduction section, we have highlighted things that have not yet been analyzed previously, and we have expanded the paragraph on weight maintenance.

We have removed some information (e.g. on body composition and biochemical parameters results) from the discussion section. We have cited previous papers similar to our study and compared the results. Despite these actions, the discussion section is still extensive, but it contains information that has rarely been analyzed in the context of weight maintenance (IPAQ, COVID-19 lockdown). The paragraph about genetic tests contains only information connected with the maintenance phase, which is also a unique value of this work. Topics covered in the discussion are very actual and we are sure that our work will be used as a reference source many times when you give us a chance to publish our paper.

Point 4: Please consider adding some figures showing the efficacy of the intervention in its different phases (compared to the control group from phase 2 until the end of follow up) on BMI, % fat or VAT, and the main metabolic parameters which the authors want to emphasize and discuss (glucose ?...others?)”

Reply 4: We appreciate the Reviewer’s suggestion. We have added a new figure about changes in BMI during the entire study in SG and CG. Due to the length of the article and the large content of the tables, we have decided to add only one figure that presented the most important indicator - BMI. We believe that based on this chart it will be easy to conclude changes in other measured parameters.

Point 5: Statistical methods : change "measurable parameters" to continuous parameters. Also "non-measurable" into categorical/ numerical ...

Reply 5: We thank the reviewer for pointing this out. We have changed the above phrases.

Point 6: Tables: a list of abbreviations should be presented at the bottom of the table. Preferably, the statistical method used for comparison within the table should be mentioned.”

Reply 6: This is the right suggestion. We have added abbreviations and statistical methods at the bottom of the table.

Point 7: Results - please add some written description on tables 5 and 6”

Reply 7: According to Reviewer's request, we have expanded a description on Table 5. In case to Table 6, a broad description was included in the original version (lines 404 – 411), which perhaps have missed the Reviewer's attention.

We thank Reviewer 1 for the constructive and insightful comments, which have helped us to substantially improve our manuscript.

Reviewer 2 Report

Abstract

Line 16. In the field of science, to include the word "demonstrate" is very bold. Please reconsider the writing of this sentence.

Line 17. Please include unit of age (years): ...22 women, aged 35.58 ± 9.85 years

There is no reference to physical activity control in the abstract. We recommend that this mention is made to help the future reader who has considered this important variable in weight loss and weight maintenance programmes.

Introduction

The introduction explains in an orderly way the current situation of obesity in the world, then explains and develops the concept of BMI, ways of measuring body components, the implications of obesity in body biochemical changes and finally approaches the problem of study from genetics and diets.

However, although it later mentions the importance of physical activity, it makes no mention of the use of intervention programmes in which it is considered and what the results have been in similar populations with the aim of weight maintenance. This should be included in the introduction, please.

Finally, it includes the hypothesis in the place that is usually dedicated to identifying the purpose of the article.

  1. Materials and Methods

2.1. Study design

2.1.1. Phase 1 of the study - weight loss process

 An imbalance is presented for weight loss through the diet in which personalised treatments (lines 139 and 142) and multiple recommendations are included, however, for physical activity only two vague recommendations are made and on which no experimental control has been presented.

2.1.2. Phase 2 of the study - post-therapeutic care

In line 174 there is only one recommendation for physical activity: 10,000 steps/day - why? This recommendation is a fallacy that originated from a Japanese pedometer salesman, who was responding to the government's intentions to get the Japanese population to exercise after the first Olympic Games in Tokyo (1964). This pedometer called "Manpo-kei" (translated from ajponese means 10,000 steps) was given away with the Sunday newspaper and the figure of 10,000 steps/day was placed as any other figure could have been placed and without being based on any scientific study (https://www.health.harvard.edu/blog/10000-steps-a-day-or-fewer-2019071117305).

Recent research has concluded that it is not only the quantity but also the intensity of daily steps that helps to maintain a healthy state.

2.2. Test group

In the abstract it states that the participants are 22 women (line 17) and in this section it states that there are 14 men and 22 women (line 207) and this is how it appears in phase 1; in phase 2 of the study there are 11 women and 7 men. This should be clarified.

2.3. Outcome measurements

Regarding the IPAQ questionnaire, it is indicated that it is filled in when visiting the dietician but it is not clear if it coincides with the same measurement moments as the rest of the variables in this section. This needs to be better explained.

  1. Results

Explain and justify why the IPAQ value at 6 months was not included.

In the tables that include t-student, please include the effect-size.

The results include men and women in the same group, not knowing, at least, if there are significant differences between them at the beginning of the experiment. Nor do we know whether at the time of the anthropometric measurements the women were menstruating, which affects their Total Body Water.

The analysis of the results shows a structure of improvement at 6 months and then a good part of the variables lose the effects produced by the follow-up at one year.

In my opinion, the vague control of physical activity as a fundamental component in the maintenance of weight in obese people has greatly weakened the possible advances that this research could have achieved.

Author Response

Reviewer 2 Changes in the manuscript are highlighted in turquoise (reviewer 1 - changes highlighted in green, editorial office - changes highlighted in yellow)

We would like to thank the Reviewer 2 for the valuable comments and efforts towards improving our manuscript. In the following, we highlight comments and suggestions from Reviewer 2 and our effort to address these concerns.

Point 1: Line 16. In the field of science, to include the word "demonstrate" is very bold. Please reconsider the writing of this sentence.”

Reply 1: We agree with the Reviewer. We have removed the word “demonstrate” from the abstract.

Point 2: Line 17. Please include unit of age (years): ...22 women, aged 35.58 ± 9.85 years”

Reply 2: This is the right remark. We have followed this suggestion.

Point 3: There is no reference to physical activity control in the abstract. We recommend that this mention is made to help the future reader who has considered this important variable in weight loss and weight maintenance programmes.”

Reply 3: This is a valuable suggestion. We have added references about the assessment of physical activity in the abstract.

Point 4: However, although it later mentions the importance of physical activity, it makes no mention of the use of intervention programmes in which it is considered and what the results have been in similar populations with the aim of weight maintenance. This should be included in the introduction, please.”

Reply 4: We appreciate the Reviewer’s suggestion. In the introduction section we have added information about physical activity and others strategies previously used in the maintenance phase. We have also extensively described the results of other long-term interventions in the discussion.

Point 5: Finally, it includes the hypothesis in the place that is usually dedicated to identifying the purpose of the article.”

Reply 5: We agree with the Reviewer. We have placed information about the purpose of the research at the end of the introduction section.

Points 6 and 13: An imbalance is presented for weight loss through the diet in which personalised treatments (lines 139 and 142) and multiple recommendations are included, however, for physical activity only two vague recommendations are made and on which no experimental control has been presented.”

„ In my opinion, the vague control of physical activity as a fundamental component in the maintenance of weight in obese people has greatly weakened the possible advances that this research could have achieved.”

Reply to points 6 and 13: We fully agree with Reviewer 2 that physical activity is one of the fundamental elements of maintaining reduced body weight. Moreover, the WHO clearly indicates that insufficient levels of physical activity are one of the two main causes of obesity. This fact also emphasizes the importance of physical activity in the process of body weight loss and maintenance.

We also agree that we did not pay to get enough attention to physical activity in our study. This is due to two main reasons:

1) In the first phase, we focused only on dietary issues. In the initial assumptions of the study, in the first stage, we did not plan to introduce any other interventions than nutritional. Of course, we encouraged participants to increase their level of physical activity "on their own" and we monitored its level twice using the IPAQ questionnaire. However, this was not a strict control. Numerous successful strategies connected with the weight loss and the maintenance process are based only on the nutritional intervention:

  • Aller EE, Larsen TM, Claus H, Lindroos AK, Kafatos A, Pfeiffer A, Martinez JA, Handjieva-Darlenska T, Kunesova M, Stender S, Saris WH, Astrup A, van Baak MA. Weight loss maintenance in overweight subjects on ad libitum diets with high or low protein content and glycemic index: the DIOGENES trial 12-month results. Int J Obes (Lond). 2014 Dec;38(12):1511-7. doi: 10.1038/ijo.2014.52.
  • Butryn, M.L., Thomas, J.G., and Lowe, M.R. 2009. Reductions in internal disinhibition during weight loss predict better weight loss maintenance. Obesity, 17(5): 1101–1103. doi:10.1038/oby.2008.646.
  • Phelan, S., Lang, W., Jordan, D., and Wing, R.R. 2009. Use of artificial sweeteners and fat-modified foods in weight loss maintainers and always-normal weight individuals. J. Obes. (Lond.), 33(10): 1183–1190. doi:10.1038/ijo.2009.147.
  • Gorin, A.A., Phelan, S., Wing, R.R., and Hill, J.O. 2004. Promoting long-term weight control: does dieting consistency matter? J. Obes. Relat. Metab. Disord. 28(2): 278–281. doi:10.1038/sj.ijo.0802550
  • Rolls, B., Roe, L., Beach, A., and Kris-Etherton, P. 2005. Provision of foods differing in energy density affects long-term weight loss. Obes. Res. 13(6): 1052–1060. doi:10.1038/oby.2005.123.

2) In the second phase of our study, we planned to include greater control of physical activity, including through regular gym classes with a physiotherapist who would tailor an exercise program for obese people. Unfortunately, the lockdown related to the COVID-19 pandemic prevented us from direct meetings and thus excluded the possibility of introducing an increase in the level of physical activity in the study group. One of the key elements of physical exercise is the correct technique of performing it. This aspect was impossible to control over the Internet. We hope that this goal can be achieved in our future research.

To respond to the Reviewer’s concern we have included information about the insufficient control of physical activity in the Limitations section.

Point 7: In line 174 there is only one recommendation for physical activity: 10,000 steps/day - why? This recommendation is a fallacy that originated from a Japanese pedometer salesman, who was responding to the government's intentions to get the Japanese population to exercise after the first Olympic Games in Tokyo (1964). This pedometer called "Manpo-kei" (translated from ajponese means 10,000 steps) was given away with the Sunday newspaper and the figure of 10,000 steps/day was placed as any other figure could have been placed and without being based on any scientific study. Recent research has concluded that it is not only the quantity but also the intensity of daily steps that helps to maintain a healthy state.”

Reply 7: We thank the reviewer for raising this critical remark. We recognize that there is no scientific evidence to support this recommendation. However, in our study, we used the Ten Top Tips method, where one of the points is about taking 10,000 steps/day. We did not want to modify this previously developed method that had been used before in research:

  • Cleo G, Glasziou P, Beller E, Isenring E, Thomas R. Habit-based interventions for weight loss maintenance in adults with overweight and obesity: a randomized controlled trial. Int J Obes (Lond). 2019 Feb;43(2):374-383. doi: 10.1038/s41366-018-0067-4.

Additionally, from a practical point of view, recommending 10,000 steps a day does not appear to be harmful. On the contrary, this goal may motivate participants to increase their daily level of physical activity.

Point 8: In the abstract, it states that the participants are 22 women (line 17) and in this section it states that there are 14 men and 22 women (line 207) and this is how it appears in phase 1; in phase 2 of the study there are 11 women and 7 men. This should be clarified.”

Reply 8: We thank the reviewer for the suggestion to clarify these values. In the abstract, we have added information about 14 men (line 17). In Phase 2 of the study, there were 36 participants: 18 of which (11 women, 7 men) were in the support group (SG), and the same number of individuals (11 women, 7 men) were in the control group (SG), so the above values are correct.

Point 9: Regarding the IPAQ questionnaire, it is indicated that it is filled in when visiting the dietician but it is not clear if it coincides with the same measurement moments as the rest of the variables in this section. This needs to be better explained.”

Reply 9: We appreciate the reviewer’s suggestion. We have added information about the time of filling the IPAQ questionnaire (line 265).

Point 10: Explain and justify why the IPAQ value at 6 months was not included.”

Reply 10: In the initial assumptions of the study, we have planned three control points: at the beginning of Phase 1 (0 month), at the beginning of Phase 2 (12th month), and the end of Phase 2 (30th month). During this time, we collected results from BIA, WC measurements, blood analyses, and IPAQ. However, the analysis of the results obtained after the end of the study prompted us to publish also the BIA results after 6 months to show more fully the changes in body composition during Phase 1. These are additional results that we did not initially plan to publish. For this reason, we do not have the IPAQ results and blood analysis after 6 months of Phase 1.

Point 11: In the tables that include t-student, please include the effect-size.”

Reply 11: This is a valuable suggestion. We have added values of effect-sizes in the Table 2. Due to the extensive Tables 3 and 4 and the limited space into them, information about the calculated values of effect-sizes is presented in a descriptive form below the tables.

Point 12: The results include men and women in the same group, not knowing, at least, if there are significant differences between them at the beginning of the experiment. Nor do we know whether at the time of the anthropometric measurements the women were menstruating, which affects their Total Body Water.”

Reply 12: We thank the Reviewer for pointing this out. Due to the small number of men in the study, we decided not to divide the respondents by sex, because the obtained results might not be reliable. There is no evidence that gender plays a significant role in the weight loss process or maintenance phase, so we believe that such a division is not essential.

The follow-up visits were monthly and usually on the same day each month to minimize the impact of different phases of the menstrual cycle on the obtained results. We have added this information (line 157).

We thank Reviewer 2 for the constructive and insightful comments, which have helped us to substantially improve our manuscript. 

Round 2

Reviewer 1 Report

The current version has been satisfactory modified.

I suggest that new figure added to the manuscript (BMI change over time) would include statistical comparison between and within groups. 

Author Response

Thank you very much for your valuable remarks allowing us to improve our manuscript. We corrected the figure as suggested.

Reviewer 2 Report

In my opinion, the suggestions, clarifications and explanations have been sufficiently answered in each of the points made in the previous review.

Many thanks to the authors for accepting and modifying what was possible from my review.

I hope that this article can be finally published congratulations to the authors.

Author Response

Thank you very much for all good words and your valuable remarks allowing us to improve our manuscript.